# Manifestations and Preconditions of Child Rights Protection—Specialists' Aggression towards Caregivers and Child in the Situation of Child Removal from the Family

**Donata Petružytė** [ID]**, Violeta Gevorgianienė** [ID]**, Jūratė Charenkova** [ID]**, Miroslavas Seniutis \*** [ID]**, Laimutė Žalimienė** [ID]**, Eglė Šumskienė** [ID] **and Lijana Gvaldaitė** [ID]

Institute of Sociology and Social Work, Vilnius University, 3 Universiteto St., LT-01513 Vilnius, Lithuania; donata.petruzyte@fsf.vu.lt (D.P.); violeta.gevorgianiene@fsf.vu.lt (V.G.); jurate.charenkova@fsf.vu.lt (J.C.); laima.zalimiene@fsf.vu.lt (L.Ž.); egle.sumskiene@fsf.vu.lt (E.Š.); lijana.gvaldaite@fsf.vu.lt (L.G.)
\* Correspondence: miroslavas.seniutis@fsf.vu.lt

**Abstract:** Numerous studies have focused on the issue of client aggression against various help professionals. Much less attention has been paid to the opposite phenomenon—the aggression of help professionals towards clients, especially aggression of child rights protection specialists (CRPS). Comparative analysis of four perspectives (CRPS, parents, children, and police officers) was performed in order to reveal the manifestations and preconditions of CRPS aggression towards parents and children during the process of removing a child from a family. The manifestations of psychological and physical CRPS aggression were alluded to by all groups of research participants. The preconditions can be classified as being related to the behavior of the child and parents, the employee's personality traits, competencies and psychological states related to a specific work situation, and institutional, inter-institutional, social, and political contexts. Based on our research results, suggestions can be made on how the procedure of removing a child from an unsafe family environment can be improved, such as by making it less harmful for children, more constructive for the whole family, and making the CRP system operate in such a manner that it does not create preconditions for CRPS to transgress the boundaries of professional relations.

**Keywords:** child rights protection; child removal from the family; helping specialists' aggression towards clients; qualitative research





## 1. Introduction

Numerous studies have focused on the issue of client aggression against various help professionals [1–9]. Much less attention has been paid to the opposite phenomenon—the aggression of help professionals towards clients. Nevertheless, published empirical data, to a greater or lesser degree, provide evidence of aggression towards clients in many areas of care: medicine, psychotherapy, and social work, among others [10–14]. Despite the fact that there are papers focused on violence in child protection and welfare services at the individual and systems levels (e.g., [15–18]), the research on workers' aggression in the situations when children are removed from their families is less common. According to Ringstad [19], workplace violence in child protective services exists, and at least one-fifth of workers report that they had perpetrated a violent act toward a client.

Therefore, we chose to examine this topical and complex issue in the national context of Lithuania, hoping that many of the insights we will present are more related to the essential features and regularities of the phenomenon than to their local national context. Our article might contribute to a better cognition of the formal and informal processes and structures in child protection services and support efforts to create a more sustainable environment for children to grow in. Therefore, the results of the study may be relevant to professionals working in the field of children's rights, individuals who have experienced a

removal of a child from the family, policymakers, and other interested citizens. The context of the study is situations when child rights protection specialists (further CRPS) come to families due to a possible violation of children's rights and make and implement a decision to take a child from an unsafe family environment. The lack of literature in this area may be an indication of the failure to acknowledge that assaults on clients occur in the sensitive area of CRP. Scarcity of research can be related as well with the fact that "victimization by professionals who are charged with providing services and who hold positions of power concerning clients' lives is at the very least unethical and, in some cases, illegal" [19] (p. 139).

In the field of protection of children's rights, the activities of specialists partially embody the aggressive function of the state, and this taboo issue acquires specific relevance. People who have challenging experiences in their families and who experience aggression from other family members are additionally traumatized by the state in the process of the removal of the child from a family [20–23]. Therefore, the methods of work of the specialists themselves in these situations require exceptional professionalism and the ability to act without additionally aggravating the atmosphere, which is also already enriched with aggression [24–28]. On the other hand, studies show that interventions in difficult family situations while performing an aggressive state function affect the professionals' well-being [29,30]. Finally, improper actions of CRPS increase public resistance and distrust of the CRP system. All of this creates communication barriers between all the actors involved in taking a child from the family and causes a great deal of stress, trauma, stigmatization, and results in exclusion (both of clients and professionals). Aggression must be controlled in order to provide effective and constructive assistance. Therefore, it is important to investigate whether this particular phenomenon exists, to what extent, in what forms, how professionals reflect it, and what explanations for aggression they can provide. Researchers recognize that this is a much-needed area of research related to ensuring the human rights of clients and the quality of services, as well as the issues of professionalism and adequate working conditions [10,31,32]. This type of research is essential for understanding what needs to be conducted in order to change and improve the CRP system [33,34]. Therefore, this article aims to reveal the manifestations and preconditions of CRPS aggression towards parents and children in the process of removing a child from the family based on the results of a complex qualitative study which allowed comparing the perspectives of four actors: (1) CRPS, (2) persons who were removed from their families as children, (3) parents from whom children were removed, and (4) police officers who accompanied CRPS in the procedure of removing children from a families.

Before discussing research methodology and results, it is necessary to define the essential concepts. It is challenging to grasp and define what aggression is, especially aggression towards clients [10], and even more difficult to define aggression in an aggression-saturated situation, which is encoded in the very nature of CRPS function and includes breach of privacy and disruption of family ties by those in power [31]. Therefore, we will apply the definition of the essential concept of aggression suggested by Yon et al. [35] for further analysis: "Overall abuse [against clients] is a single, or repeated act, or lack of appropriate action, occurring within any relationship [...] which causes harm or distress to an [client]" (p. 150). Thus, in this study, when analyzing data, we looked for the elements of CRPS (in)action mentioned in the narratives of each group of study participants, which caused stress or harm to parents and children, and treated them as manifestations of aggression. In the empirical material, while searching for preconditions, we sought either direct explanations from study participants about the reasons for such CRPS (in)action or hints about the indirect conditions that heightened CRPS stress in the work atmosphere, assuming that this increases the likelihood of CRPS (in)action mentioned earlier. Relying on an analysis of the most relevant research regarding service providers' aggression towards clients in a variety of settings, a conceptual framework and analytical tool to obtain comprehensive understanding of the phenomenon from empirical data was created [6,14,19,33,36].

## 2. Methodology

The issue of aggression in help relationships is very sensitive. Often, publicity for this issue in the media focuses on the accusations of professionals, highlighting their lack of competence or inability to work with clients. Thus, although some helping professionals may be reluctant to discuss topics of aggression because they really do not see that they are doing anything wrong, not surprisingly, some help professionals avoid openly sharing their experiences because of the media circus and public judgments. Due to their traumatic and stigmatizing experiences of aggressive behavior by help professionals, clients also avoid giving their accounts on this for fear of being misunderstood, ridiculed, or later harmed [11,37]. Thus, research in this area faces methodological problems, such as the reliability of information [10]. According to Stanley et al. [10], when interviewing employees, there is a high probability that they will not fully disclose their aggressive actions towards clients. On the other hand, clients' recall of such situations may also be incomplete. Therefore, in order to obtain the most reliable information possible, a complex qualitative research method was chosen—a semi-structured interview with four different groups of participants:

(a) Specialists from the CRP agencies (hereinafter in the article, the term "specialists" will be used to refer to this group);
(b) Adults who have personally experienced being removed from their families as children (hereinafter referred to as "children");
(c) Parents whose children were removed (hereinafter referred to as "parents");
(d) Police workers who participated in the procedure of removing a child/children from the family (hereinafter referred to as "police officers").

**Research participants and setting.** A total of ten semi-structured interviews with child protection workers, each lasting from 28 to 80 min, were conducted in children's rights protection offices in 2020. The sample of informants was based on the list of child protection workers, which was obtained from public websites of the 12 regional child protection agencies in Lithuania. The interviews were conducted with employees working with families in various regions (large cities, smaller district towns, and rural areas). Based on the job description, two groups of informants were selected from the staff list: on-call staff and chief specialists who were assigned to go to families on a day-to-night basis due to a threat to a child's safety. Before agreeing on an interview by telephone with the employees performing the aforementioned functions, the State Child Rights Protection and Adoption Service's written consent for the interviews was obtained. The study included three male and seven female specialists with work experience ranging from 0.5 to 30 years and having different professional backgrounds (social work, education, law, management, and public administration).

In addition, ten semi-structured interviews with police officers who had the experience of accompanying child protection workers in the procedure of child removal from the family were conducted in police offices during 2020. The interviews lasted from 12 to 48 min. The police department compiled a list of police officers with child removal experience and provided the researchers with their contacts. From this list, the researchers selected the future participants of the study in such a manner that they would represent different regions of Lithuania and settlements of different sizes. This was also performed to reflect the two groups of officers involved in the child removal process: patrols and response units. The sample involved four men and six women with a length of service ranging from 1 to 12 years.

Ten semi-structured interviews with adults who were removed from their biological families during childhood were also conducted in 2020. Each of these interviews lasted from 23 to 90 min. In the same year, additional ten semi-structured interviews with parents who had experiences of child removal from their family were conducted. The interviews lasted from 36 to 68 min. Criterion and snowball selections were used for these two types of recruitment of informants.

At the start of the study, the researchers realized that finding study participants and persuading them to participate was difficult enough due to the particular stigma surrounding the issue. Therefore, the strategy chosen was to initially disseminate information about the study to various organizations: family associations, crisis centers, temporary housing, organizations providing social services for parents or children, and childcare institutions. When clients agreed to participate in the study, their contacts were passed onto the researchers. The interviews with parents were conducted by the members of the research team who had an educational background in social work and psychotherapy. Due to the complicated accessibility of the study participants, they were recruited regardless of geographical aspects. Some adults who were removed from their biological families during childhood agreed to meet with the researchers and later changed their minds. Some interviews were divided into two parts (i.e., conducted with a break) or required a preparatory or concluding interview with the researcher due to intense and complex emotional experiences in an effort to recall childhood experiences. There were five women and five men in this sample, with an average age of 20 years (min—16; max—28). In addition, the average age of parents who had experienced child removal from their family was about 37 years (min—21; max—55). The sample of parents included seven women and three men.

**Data Collection and Analysis.** In order to reveal participants' experiences related to the procedure of taking a child from the family, semi-structured interviews [38,39] were conducted, which involved four groups of informants mentioned earlier. The interviews were audio recorded and, subsequently, transcribed verbatim. Based on the interviews' transcriptions, data analysis was conducted using a mixed deductive-inductive qualitative content analysis approach [40–43]. Firstly, the transcriptions were reviewed for segments related to the main categories of manifestations and preconditions of CRPS aggression (derived deductively). Afterward, the transcriptions were coded using MaxQDA 2020 software for qualitative research. Since data were collected from four different groups of informants, a further analysis was conducted separately and then compared, focusing on emerging sub-categories. Examples of inductively derived sub-categories from parent interviews are as follows: (*) incomprehensible/complicated erroneous explanations related to the reasons for the child removal; (*) direct, rude, cold, and indifferent behavior of CRPS that puts much stress on many family members; and (*) mocking, discriminatory, ridiculous behavior, or the language of CRPS. In the final stage, researchers worked through the coded segments once again and, in parallel, reformulated initial sub-categories by transforming them into a united system of categories and sub-categories.

**Research ethics.** Ethical approval for the research was granted by the Institute of Sociology and Social Work. Permission to interview child rights protection workers was obtained from the State Child Rights Protection and Adoption Service under the Ministry of Social Security and Labor. Permission to interview police officers was obtained from the Police Department under the Ministry of the Interior of the Republic of Lithuania. Before the interviews, an informed consent was obtained from all participants who agreed to participate in the study.

## 3. Research Results

### 3.1. Manifestations of CRPS Aggression: Children's Perspective

What has caused a significant amount of stress for many children is the suddenness, unexpectedness of, and unpreparedness for the intervention. One research participant shared the following:

> Furthermore, right away, you know, it felt like a direct stabbing of the heart with a stick, stabbed my heart directly with a stick, and they take away from your mother, that is how it felt, at least to me [V1].

According to the children, specialists should "not take those children so suddenly". They should "talk to those children at first" [V3] and prepare them and not take the approach where someone "comes and you do not know what is going on here" [V7] and "strangers take you somewhere, this is the most striking" [V6].

What concerns CRPS communication is that children remembered that a great deal of stress was caused by insufficient information: "My uncle is taking me somewhere, and some unfamiliar woman is sitting there. [...] I could not understand where I was being taken" [V10]. From a children's perspective, it can be argued that often professionals do not provide essential information to a child in comprehensible form. In some cases, it seems that the child's confusion and the lack of grasp of what is happening and what the future holds for him/her become so deep that it lasts long after the removal.

Another problem that many children mentioned in their interviews was misleading or unrealistic information provided on the part of professionals or, in children's words, a pack of lies: "And when you take children away from the families, you do not need professionalism here, you just need a good tongue so that you can lie a bit, sometimes until reaching the car. That is all, nothing more is needed" [V10]. They were lied to about the situation in the family and the reasons for taking them, about where they go, for how long, how good was this new place (unrealistic narration of the place of care, silencing its shortcomings, or even beautifying it with "non-existent things" [V10]). The consequences of the removal from a family last a lifetime. One informant, who was deceived from his hiding place and taken from the family, commented that the lies of the specialists were "the worst thing that can happen, because, in that sense, your confidence is immediately halved" [V2].

The children also disapprove that the professionals did not consider their or their loved ones' opinions when making decisions: "Everything was done straightforwardly, without even asking us what made it happen here, [...] asking nothing and simply taking it away" [V4]. From the perspective of children, the "independent" decisions of professionals are not necessarily wholly correct. As one informant puts it: "I will do that and that, how I think is better, and everyone is content, everyone is happy, but that child will not be happy at all" [V5].

Regarding CRPS behavior, some children alleged that the severity of CRPS particularly stressed them, and their formality in a very sensitive situation towards them determined their future life. They were also concerned about the roughness and coldness of the manner professionals addressed their parents. In this respect, the behavior of police officers, which is sometimes particularly intimidating to children, has a special role to play. One study participant said that if she had to work as a child specialist, "I do not know if I would go with the police" [V10].

Finally, children mentioned specific CRPS physical actions of removing them that have caused them great stress and harm. One participant shared the following: "they just tried to physically pull me, rob me from under the bed, I kept screaming, screaming and saying I was waiting for my mom" [V7]. Other informants wondered why CRPS did not choose different and less radical methods: "it seems it could have been done through psychologists, well, so as not just to tear children from those parents" [V4].

*3.2. Manifestations of CRPS Aggression: Parents' Perspective*

Regarding the very nature of the intervention, many parents expressed their dissatisfaction with the suddenness and unexpectedness of CRPS intervention, its speed, and shortness of the process.

In the retrospective assessment of CRPS behavior, it was the insufficient specialists' communication that caused parents the most stress and anxiety during the removal and anger after that procedure: "I am most angry that they did not explain everything to me at that time" [S2]. According to the parents, CRPS communication during the removal was inadequate and insufficient. CRPS failed to explain the reasons for removing the child and how to obtain the child once more. One study participant said that CRPS did not explain why the child was being removed and ignored her for three days when she arrived at the CRP office attempting to obtain her child. The insufficient explanation received from the CRPS not only increases stress experienced by parents in the child removal situation but also causes their intense resistance to the CRPS when they try to

separate the child physically: "[CRPS] says: 'I take [the child] to the foster parents'. I ask: 'where?' She answers: 'Not your business. You will find out everything'. And I started to scream. Hooked up to my children" [S7]. In addition, parents do not usually have enough information about the reasons for taking their child and were inclined to question the legality of the decision to remove the child.

Regarding CRPS behavior, it can be observed that the stress for many family members was also caused by directives and rough, cold, and indifferent CRPS behavior during the removal of a child from the family. For example, one study participant said:

> They said, "go where you want," and added, "without a child". And then I just froze and remained speechless. Moreover, And even after I asked [CRPS] "why?", she started talking to her colleague there, and did not pay any attention to me, whether I would cry or did something else [S9].

Reflecting on what CRPS could have performed differently to make the removal of a childless less stressful, parents often expressed a desire for professionals to reconsider their intervention methods to "take children more gently from families [...] so that there is no stress for either parent, or children" [S11].

Reflecting on the experience of child removal, some parents emphasized the pressure of CRPS to agree with their decision regarding child removal. One study participant said that she faced this pressure at a meeting that decided on the continued custody of her children. Some research participants shared that they were receiving threats from CRPS during the removal: "If you resist, we will take you for treatment" [S6]. Other parents also shared that CRPS were against them; they did not want to help the family solve their social or psychological problems but only to "take away" the children as fast as possible. Without understanding the motives for CRPS behavior (i.e., removing a child) and feeling compelled to choose a solution that did not satisfy them, the parents felt deceived, and this sense of deception encouraged them to question the legitimacy of both the CRPS decision and the separation of the child from the family. Furthermore, persistent resentment often complicated the subsequent help process, as the parents opposed the intervention of other specialists and did not trust their efforts to help sincerely, and some parents locked themselves in and embraced self-destructive intentions.

As some of the study participants reported that they had even experienced physical CRPS actions during the child's removal, in other words, aggression:

> Because the exact moment when children are snatched from my hands ... That moment was like a nightmare. Such a feeling that something from the chest is ripping the heart. When those kids were removed, they took away a part of me. It's a horror <...> And I am screaming. After grabbing those kids. She caught them, one by one, and took them out [S15].

Child removal situations were susceptible to all participants in the process. The research participants described the moments where CRPS physically separated the child from the parents as highly traumatic events and with long-term negative consequences for both children and parents' psychological well-being.

### 3.3. Manifestations of CRPS Aggression: Police Officers' Perspective

Accounts of some of the police officers reveal that the behavior of CRPS often seems to lack a sincere desire to help the family and the child. Informants attribute this to CRPS's reluctance to delve into the situation: "They do not pay attention. All I can say about their communication and culture is that there are situations when you want humanity from them, which I very rarely notice in their behavior" [P4]. As police officers put it, sometimes it seems that the CRPS formally perform their job: "there are situations when they do their work reluctantly,that is, usually they are not happy that they have had to come" [P4]. Sensitivity is especially lacking in situations where it was necessary to physically separate a child from his parents. Such situations are also sensitively experienced by officials:

> Whoever is more sensitive, feels pain. The girl is crying; she needs to be torn from her mother, almost by force; her mother is also crying; (she is sober). Moreover, you have such an ambiguous feeling, that you are taking a child from a mother's embrace [P2].

The use of physical actions both by CRPS and by police officers in such situations was perceived exceptionally negatively. Although officers can use physical force legally, they were more inclined to try to resolve the conflict verbally:

> There are also situations where minimal force needs to be used, even in the example I mentioned before, from the mother's embrace, using a coercive force, well, you really can not take that baby, you better start talking explaining that it hurts the child, you know. And you try to reduce the tension of that situation to a minimum... In that case, you should not use that physical force... Still, you must make that contact with that child yourself, so that he is calmer at the time, so that he has at least some kind of trust or contact with you. He/she could meet the professionals and travel further [P2].

*3.4. Manifestations of CRPS Aggression: CRPS' Perspective*

Even though CRPS spoke relatively little about their aggressive behavior while communicating with clients, they acknowledge that their communication with clients is sometimes inappropriate. One of the manifestations of aggression that they mention is speaking in a raised tone of voice. In addition, the informants realize that their own communications can provoke clients: "If you come somehow demonstratively, with aggression, with some uplifting tone, that opposition will be twofold, especially if the person is intoxicated" [VD2]. Another aspect is the use of the incorrect vocabulary. In her account, one informant says that when talking to a client, it is essential to use such a vocabulary that is readily understandable to that person: "I just said to that woman... I go down to her level, and I even talk sometimes with jargon, really" [VD2].

When it comes to possible cases of clients' physical abuse, CRPS are more likely to talk about the situations that have occurred in the practice of their colleagues than to disclose their personal professional experience. One of the informants referred to a case where, in response to the parents' aggressive behavior, CRPS engaged in a fight: "it was an outbreak of fisticuffs" [VD3]. The informants also mentioned cases when they had to take a child from an unsafe environment, which they interpreted as a necessity: "when the child is taken from the mother from that embrace, somehow in that situation you do not feel empathy, because you see how badly the mother behaves, how bad everything is there" [VD1]. In his account, another informant talked about some interventions as organized actions in which police officers were also involved:

> How to remove that child when she is holding him in her hands, then you agree with the officers: you hold the mother and we remove the child. We had to grapple to take that child from his mother [VD1].

Immediate child removal using physical force is a relatively rare practice. Other similar CRPS strategies complement the range of manifestations of physical aggression against the client. Commenting on this issue, two informants mention several cases of taking a child from the family, which, from the parents' point of view, could be equated with "kidnapping":

> Mother was hiding herself and her children, and we managed to take two children, get into a car, and sit inside the locked car, the children were crying [...]
> In fact, there were cases when we have had to press the gas and run away quickly [VD7].

In a similar vein, research participants mentioned that, in their professional practice, they have had to take steps to enter into the private family space without parental consent, deal with family belongings arbitrarily, and even damage family property: "there have

been situations where we have tried to remove the window ourselves [in order to save a choking child]" [VD4].

### 3.5. Preconditions of CRPS Aggression towards Clients: Children's Perspective

**CRPS personal qualities.** From the children's point of view, preconditions for aggression are created when professionals lack personal qualities such as kindness, openness, sincerity, sensitivity, and empathy. One research participant said the following:

> They have to be as crazy as the children themselves. They do not have to be afraid to get dirty; they do not have to be afraid to fall, or be afraid to play with the kids, or be there like them. Well, they have to be friends with that child, they have to be adults, and they have to be role models [V5].

In the absence of such qualities, it is difficult to remove a child from an unsafe environment without causing him additional strain.

**CRPS competencies.** Several study participants noted that the high level of stress they experienced with the foster family was related to the lack of professional competencies of the specialists. Some emphasized the lack of knowledge in child psychology and psychology of communication. Others observed that specialists were inexperienced, while some said they simply could not connect with them, i.e., listen to them and explain what is happening understandably.

**CRPS motivation.** Some children linked their difficult experiences during the removal to the formal attitude of CRPS. Some, and some believed that it was performed based on material gain: "it feels as if they work just because they get money for it [in a sense, that some research participants believe that CRPS receive an additional bonus every time when they decide to take a child from his or her family]" [V7].

**Institutional preconditions.** In interpreting the data set as a whole, we attributed some of the children's statements to institutional preconditions due to their content, although the children themselves talked about them more as of the workers' characteristics. Study participants stated that the decisions made by CRPS during the single visit to the family and, consequently, insufficient immersion in the situation were too abrupt. Some children who were taken at older ages said that CRPS did not try to involve them more in the decision to remain or not in the family. Importantly, children associated this with CRPS's reckless action, laziness, irresponsibility, superficiality, and very indifferent relationship with their work in a situation that children remembered as truly fateful for them. By interpreting these thoughts of the research participants, we observe the preconditions of such behavior in institutional procedures, which do not allow for a "slower" operation of CRPS and greater involvement of clients in the decision-making process. In the absence of these institutional possibilities, there is a growing likelihood of such CRPS (in)action that children experience as causing significant stress and harm to them.

### 3.6. Preconditions of CRPS Aggression towards Clients: Parents' Perspective

**CRPS personal qualities.** The prerequisite for the improper or aggressive actions of CRPS, identified in the parents' interviews, was the lack of personal maturity (which the research participants associated with bringing up their children). The experience of raising children, according to the informants, would help CRPS to perform their job better and to understand the perspective of both parents and children. In their accounts, some of the informants attributed the lack of maturity of CRPS to their indifference and insensitivity. While other informants tended to link the way CRPS worked to the "difficulty" of their personalities. In their opinion, a CRP specialist "must be a good person"; meanwhile, it was difficult for study participants to communicate with these specialists, and their communication methods were perceived as frightening and worrisome. This was especially felt when informants contrasted the experience of communicating with CRPS with other help professionals.

**CRPS competencies.** According to the parents, CRPS's aggressive behavior during the child's removal is related to the lack of specialists' competencies, relevant knowledge,

and education: "maybe they have not been trained yet" [S6]. The interviews highlighted the importance of empathetic communication skills with a sincere desire to help the family solve existing problems and to take back their children. It would have been easier for them to accept a child removal situation if CRPS had communicated what actions parents had take further in simple and plain language.

As stated by the parents, good communication skills are essential when communicating with children, and CRPS seems to be lacking knowledge on how to separate children from their parents in such a manner so as not to cause stress, shock, or trauma. According to the parents, improper communication with the child increases the feeling of not catching up in the situation and, thus, intensifies the trauma experienced during the removal.

**CRPS motivation.** As noted by the parents, CRPS misbehavior is due to their inadequate motivation to perform the work, which is so difficult in an ethical sense. Some parents were convinced that CRPS "only watch how children are taken away" [S4]. The parents also emphasized that specialists lacked insight into the situation, and more time to observe and evaluate the family was needed. Insincere dedication to work in parents' interviews was associated with insufficient CRPS attention to parents' motivation to change and their failure to evaluate their efforts to solve problems and to delve into their perspective. Parents were particularly prone to contrast CRPS's efforts to help the family and preserve its integrity with the efforts to split the family and to punish it. In addition, some informants believed that, in many cases, child removal from the family was not determined by a sincere desire to help or a real threat in the family environment but by CRPS's financial interest: "when children are picked up, they receive bonuses" [S9]. In some cases, in smaller towns where CRPS and parents often knew each other closely, informants claimed that their children were removed not for objective reasons but for personal revenge.

**Institutional preconditions.** Prerequisites of CRPS aggression were associated with the organizational culture. The misbehavior of specialists was especially felt in the smaller settlements. For research participants, it was "obvious" that CRPS worked better in the capital and large cities of the country. In their accounts, some informants also mentioned certain features of the work organization of the specialists that could become a precondition for aggression in the workplace. For example, one research participant emphasized the need to update family information stored in the CRPS database. The problem with the practice of information storage is twofold: on the one hand, it encompasses historical data that may no longer be relevant, and the information about the families in the database is not made available to the parents themselves on the other hand.

**Political preconditions.** Some interviewees mentioned that CRPS misconduct emerged due to an inadequate legal framework, which was called the "non-employees legislate" [S7]. According to one informant, this assumption helps to reconcile CRPS intervention, which is not always acceptable. As a result of this attitude, some study participants were inclined not to believe that the CRP system would change for the better in the future or that the specialist intervention would become more effective.

### 3.7. Preconditions of CRPS Aggression towards Clients: Police Officers' Perspective

**Client behavior.** From the point of view of police officers, the preconditions for the aggressive behavior of CRPS can be created by the behavior of family members such as psychological aggression (verbal hostility and swearing) and physical aggression (use of force and dog harassment) towards CRPS and officials. Such behavior is particularly exacerbated by intoxication. On the other hand, according to police officers, "sometimes such tension caused unnecessarily [by the specialists or officials themselves] provokes that aggression" [P4].

**CRPS personal qualities.** As a few officials suggested, these assumptions may be related to CRPS personality traits, such as indifference, irresponsibility ("there are people who behave carelessly" [P5]), or their temporary emotional states that hinder the smooth performance of work duties ("they are also people, they also may have lack of sleep, and they do have problems" [P8]).

**Situational preconditions.** Some police officers emphasized that "usually all situations involving the rights of a child, the deprivation of a child, the removal of a child from the family, they are all tense" [P10], causing many strong emotions that sometimes required good skills to suppress them ("not breaking down" [P8]), even from officials and professionals who have much experience. In some situations, there is a particular necessity to ensure the safety of children, so the professional behavior of CRPS is essential.

**Institutional preconditions.** From the accounts of the officials, we also learned that the direct work with clients was overshadowed by filling out documents, and informants especially stressed with high workloads prevented good quality in performing all duties and caused tension in the CRPS. This is especially true in areas with a small population (due to which they are served by one CRPS team) and long distances between settlements (it is time consuming to move from one call to another if they occur in succession).

**Social preconditions.** Officials have pointed out that the specific context of CRPS work is created, and additional tensions are caused by the media circus and negative public attitudes:

> On the other hand, despite the fact that they do not have that public support as professionals for children's rights, maybe that is why it causes them to behave like that, but often it depends on their behavior whether the public will support them. After all, society is the people you remove those children from, and they are also part of that society [P4].

However, from the perspective of the police officers involved in the research, CRPS should not use any physical aggression since there is the police force. CRPS either arrives in a safe and controlled situation or, if they arrive on their own, should call the police and ask them to use disciplinary measures or even physical force to manage difficult situations in the family. In other words, there can be no formally justifiable preconditions for CRPS aggression.

*3.8. Preconditions of CRPS Aggression towards Clients: CRPS' Perspective*

**Client behavior.** According to the research participants, one of the preconditions for their aggressive behavior in situations in which they receive verbal aggression from clients and sometimes even serious threats is as follows: "We were afraid to walk the streets alone because they were threatening us, claiming that they will kill us, bury us, you know" [VD4]. Some of the informants mentioned that when they came to the family, they had to face aggressive physical actions of their parents, such as pushing, pouring, scratching, cutting, and throwing objects:

> There were situations when they wanted to smash the car, to puncture the tires. We were also fleeing, and locking up with the kids in the car, while another employee was picking up the other kids. Indeed, we experienced many situations and were going through a lot of stress [VD7].

Specialists attributed aggression primarily to clients' intoxication and sometimes to mental illness: "There are different parents who may not be intoxicated but have certain disorders, or who have an exacerbation of a disease" [VD5]. Often, study participants also remembered the children and their feelings during the removal and admitted that children felt anxious, angry, hateful, often frightened, and hostile: "In the beginning, children are frightened" and "obviously they resist the most" [VD2] and "it happens, that sometimes teenagers escape, we fail to take them, they disappear" [VD6].

Informants partially understand such client behavior and acknowledge that "[CRPS] are still outsiders, interfering in their personal lives; [they] are defending that territory" [VD3]. In their view, it is understandable that the children who are to be removed show hostility towards professionals: "Some strangers have come and are trying to explain something to you here and now" [VD6]. Finally, professionals admit that their own arrogant, demonstrative, and aggressive behavior when they come to a family can provoke client aggression and emphasized the importance of professionalism in managing these

complicated situations. In addition, informants sometimes mentioned different types of cases that created preconditions for aggressive specialists' behavior, i.e., when faced with a father's lie, efforts to hide children or even to escape, or simply not to let professionals into their house.

**CRPS motivation.** CRPS's approach to the work at hand can contribute to the expression of aggression towards clients. Most informants claim to work out of vocation for the well-being of children. This contrasts with the formal, functional relationship with their work as revealed by other informants ("It is my job to do that task, and I am doing it. And that is it" [VD8]) and the objectifying relationship with clients.

**CRPS competencies.** CRPS professional competencies such as the ability to accept the client's emotionality and adequately express their own emotions concerning the client are essential for probable aggressive behavior scenarios:

> Of course, that patience is needed, because it happens that you travel to the family, and they start calling you there in every way, and they say "you probably do not have kids yourself, you do not know what it is, or you have kids, and your kids are also of some bad kind"... and you try to keep silent the next time and you try not even to pay attention [VD3].

Some informants reported a breakthrough after gaining work experience that initiated professional activity without emotional engagement when employees' sensitivities were replaced by a formal, emotionally focused, and information-seeking relationship with the client. Other participants in the study have pointed out that the lack of experience in dealing with uncertain circumstances in law and other work procedures also becomes a significant source of tension, especially for the newly employed CRPS:

> Probably what was the most difficult since we just started to work... the night shift, [...] we were thrown in, and allowed to do what we wanted, in a sense, just act according to the law. However, not everything is regulated by law. Well, even when we were called to that event, first of all, we had little experience, second, it caused confusion, because we did not know where to call, we started googling, and probably it also hurts your ego, that you are coming and you do not know what to do [VD7].

**Situational preconditions.** According to the informants, removing a child from the family causes a great deal of solid and difficult experiences ("These emotions work, you experience them, you bring them back to the office and sometimes home" [VD1]) and often contradictory feelings ("Because of that pity, compassion, you know, in parallel, very often goes anger" [VD3]). Among all the other emotions as preconditions, specialists especially distinguished the most common, which was stress, as well as the often felt feelings of uncertainty and ignorance, strong doubts in decision making, and fear of risk and mistakes:

> Those doubts about why we had taken a child away, in a sense, I would not like to have any doubts that we might not have had to remove that child, because then we face a lot of problems [VD7];

> A minor was killed. It was my child there, in the sense that I took her out of the crisis center for the first time. [...] a week ago, I talked with her, and I was really in much pain, and then I was in those doubts if I did everything I had. [...] I have suffered. [...] It is hard for me to talk about it now [VD4].

**Institutional assumptions.** The help process carried out by the CRPS is fragmented, which poses additional challenges to CRPS. First of all, the informants talked about the challenge of coordinating actions between the employees of different institutions because the most important decisions related to the child's well-being belong to the specialists of the CRPS. Second, study participants understand that their interventions, especially those related to taking a child from an unsafe environment, are not an extension of the work performed with a particular family in the long run. As one informant (CRPS) said: "Some strangers have come here and are trying to explain to you something now, which in turn

does not help CRPS to gain the trust of their clients" [VD3]. Third, the staff is recruited around the clock to ensure specific functions, such as on-call duty. In their accounts, the informants mentioned having concerns that such work organization does not help taking full responsibility in terms of working with a client.

The additional workload of CRPS compromises the situation, and the time spent working is increased by paper-based client data collection tools: "The questions are repetitive, but they are similar, almost the same within the same, and you have to rewrite that information several times" [VD6]. Informants mentioned that they were under pressure from managers to handle large volumes of documentation promptly in each case. This prompted them to prioritize administrative work over the welfare of the child.

The decision-making tools applied by CRPS, in the view of the informants, do not correspond to reality. They are not sufficiently individualized and can contribute to the generation of formal decisions or even prevent specialists from making the necessary decisions in the best interests of the child ("Sometimes I feel I have to remove that child, but when it comes to writing there is nothing to write" [VD6]). The information about the client that is recorded by the CRPS in the questionnaires and then transferred to the information system allows forming a narrow understanding of the client. In this sense, the employee must make professional decisions without having sufficiently detailed data about the client or forces them to go beyond standardized work procedures. In some cases, when removing a child from an unsafe environment, the CRPS faces the challenge of finding a temporary accommodation facility or guardian for him or her: "This is another part of the stress we experience at work. We do not know where to replace the child. There is a lack of those caregivers" [VD4].

Many informants also stated that there is a real lack of security at work, and they have to take care of that security themselves. They have claimed that the CRPS faces not only psychological and physical aggression on the part of the clients, but they are also concerned about the risk of communicable diseases in the workplace: "The risk of communicable diseases is high when you go those places are with tuberculosis, an open form… that contact is indeed common" [VD1].

The inadequacy of workloads, time, and material resources for real situations creates particular preconditions for inappropriate behavior of CRPS. One research participant shared that as soon as you receive a call, the first thought is always, "How will you keep up?" [VD1]. Another research participant observed the following: "Huge workloads, you know, and a lack of staff. Well, these vacancies are not filled in for years, as many people come as many leave, or maybe not, because they do not uplift the load, the whole emotional state" [VD6]. Another specialist shared her experience when the number of staff was not adequate relative to the workload:

> There was just a lot of fatigue. You could not focus on that case, on the family, because then there was so much chaos in your mind, because you know it is not the end here because you already have to do it here, go there, and write something there. Then it is very, very detrimental to your professionalism and the quality of your work because you cannot just focus on so much of everything. […] And when you do not manage to do something, it looks like you are already worthless. You are not welcomed to work here anymore [VD10].

Another research participant mentioned that a lack of material resources, such as the number of vehicles available, hindered the efficient use of time. Research participants also admitted that they sometimes have to work unaccounted for and unpaid overtime.

From the accounts of the research participants, we also learned that a large part of them expressed a desire to unburden themselves to someone and to receive support: "That emotional experience… in such cases, psychologists and psychotherapists would be unequivocally needed when one experienced such things" [VD7]. The lack of psychological support for specialists in the Lithuanian CRP system causes a high level of stress and increases the probability of their unprofessional behavior. In addition, professionals acknowledge the lack of training on topics relevant to their work.

Research participants also talked about the fact that teamwork of professionals is essential when it comes to removing a child. When there is good alignment in the team, the roles are divided according to the competencies of the team members, and the co-workers provide psychological support for each other (thus, compensating for the shortcomings of the system): "If you feel bad, you say it, you ventilate it" [VD7]. It can be argued that when there is no such alignment, the disagreement or the tension of the relationship between co-workers can become a precondition for manifestations of aggression.

Some research participants alleged that a lack of local leadership support also increased the experience of tension ("Reports and accusations only" [VD4] and "Sometimes you feel like at school, that you are always tested very heavily, you are not given any freedom of expression" [VD1]). Many specialists emphasized the distance created by the central government: "[there is] such distancing of the "top" from us, ordinary working people, who do that "dirty" work" [VD3].

**Preconditions related to inter-institutional cooperation.** The research participants revealed that they were closely connected with and dependent on other institutions and acknowledged that they were facing difficulties, which in turn created additional obstacles and increased the complexity and stress of their work (e.g., cooperation with the police or doctors is not always smooth, and the childcare system is developed insufficiently).

**Political preconditions.** In their accounts, specialists mentioned that the unstable, constantly changing legal regulations in the field of CRP also hinder their work: "I feel angry sometimes, due to those bureaucratic procedures, which often change and may not be perfect" [VD8]. The lack of integrated services for a family also raised tensions in CRPS: "Anger is sometimes caused by the lack of opportunities to help the family, services which may not be received and cannot be guaranteed" [VD3].

**Social preconditions.** Research participants often complained that they had to face negative media attention when writing about inappropriate actions of professionals. Specialists have also acknowledged that they are outraged by the fact that "the media, the press, social networks and [information] are distorted, untrue" [VD6]. Employees themselves are prohibited from speaking in public and defending themselves: "We are severely restricted by personal data protection laws, both for adults and especially for children, we really cannot come and say out loud in front of the cameras what everyone knows about the fact" [VD1].

Moreover, a large number of specialists said that society underestimates them as professionals:

> Usually, we are the people who are not good. Moreover, you can have whatever personal qualities you want in that function, but we are in a particular position. If you are a teacher, you will still be respected by everyone [...]. If you are here, [...] that is another position [VD3].

Most CRPSs shared the tension of experiencing stigma:

> When you drive a car and have the "Children's Rights" logo when they try to push you off the road so that you hit somewhere when they spit on your car, when your neighbors know where you work and do not say "hi" to you [VD6].

## 4. Discussion

### 4.1. Summary of the Most Important Findings and Interpretations

Our comprehensive qualitative research has shed light on the sensitive, complex, and little-studied phenomenon of aggression by support professionals towards clients, allowing us to approach it from four very different perspectives. What does the children's view give us? Children are the ones who are removed, and they are "rescued" [44–47], but they are also the ones who very vividly experience aggression. It allows us to understand what traumatizes and injures children during the removal process. In public discourse, parents are often treated as "bad guys" [48–52] from whom children are taken because they are incapable of providing proper care for their children. From the perspective of parents, in taking the children, they see themselves not only as abusers but also as human

beings. In their interviews, the parents not only expressed concern for themselves and their children, they blamed CRPS, but they tried to understand their actions in a broader context at the same time. These two perspectives pointed particularly clearly to mistakes that no one professional has probably avoided in their practice. In this sense, CRPS is no exception [14,19]. They can also make mistakes, and some of them openly admitted this during the interview. In this study, police officers served as impartial witnesses [53,54] who tended to expose manifestations of aggression in a different manner than CRPS (who naturally tended to adopt a defensive stance and diminish manifestations of their aggression) and unlike children or parents (who tended to highlight some cases of CRPS actions owing to the coercive nature of the CRPS interventions).

However, a comparison of the four perspectives reveals general trends. We can conclude that the manifestations of CRPS aggression were primarily mentioned in the accounts by children and parents, which is understandable because, in cases when the removal was a harrowing experience, the moments of stress and pain were the ones that were the most ingrained in their memory. Meanwhile, CRPSs themselves, more than the other three groups of research participants, elaborated the aspect of preconditions for aggressive behavior, which is understandable because they analyzed the circumstances that created prerequisites for professional mistakes or that resulted in non-standard solutions. This can also be attributed to their efforts to reflect on their professional experience in order to improve it. On the other hand, focusing on the assumptions of their aggressive behavior can be seen as an attempt to justify themselves and to reduce their responsibilities.

All groups of research participants alluded to the manifestations of CRPS aggression (both psychological and physical) towards clients (see Table 1). Children usually treated the manifestations of aggression of specialists during the removal as a pointless manifestation of their lousy personality traits, formal work motivation, or lack of competencies. The majority of the parents in the study held similar views. However, parents also mentioned the macro factors that probably had determined the actions of professionals. They pointed out that specialists acted not so much independently but rather passively sunken into the imperfect system, which was formed by the political and institutional context, and carried out the tasks set by this system in with respect to the ways they were imposed. Police officers attributed manifestations of CRPS aggression either to their professional misconduct arising due to their careless relationship with their work or the need to perform their duties at any cost by removing a child from an unsafe environment. Specialists themselves acknowledged that sometimes formal and rude behavior and unexpected "outbursts" are manifestations of their unprofessionalism, which is sometimes due to the complex nature of their work and the flawed institutional, interinstitutional, political, and social context. However, they also pointed out that in some cases, the circumstances were such that it was impossible to simultaneously perform their work functions and to protect the participants in the process without some other elements of aggressive (sometimes even physical) behavior. In addition, they claimed to be confronted with situations where aggression played a self-defense function when interacting with aggressive clients.

**Table 1.** Manifestations and preconditions of child protection specialists' aggression towards clients from the perspective of four groups of the study participants

| Groups of Research Participants | Manifestations of CRPS Aggression towards Clients | | | | Preconditions of CRPS Aggression towards Clients | | | | | | | | |
|---|---|---|---|---|---|---|---|---|---|---|---|---|---|
| | Nature of intervention | CRPS communication | CRPS behavior | CRPS physical actions | Client behavior | CRPS personal characteristics | CRPS competencies | CRPS intentions, motivation | Situational preconditions | Institutional preconditions | Inter-institutional cooperation | Political preconditions | Social preconditions |
| **Children** | + | + | + | + | | + | + | + | | + | | | |
| **Parents** | + | + | + | + | | + | + | + | | + | | + | |
| **Police officers** | | | + | + | + | + | | | + | + | + | | + |
| **Specialists** | | + | | + | + | | + | + | + | + | + | + | + |

## 4.2. Limitations

The findings from the study should be considered within the limitations of the methodology. The four samples were small, and a larger sample size would have made this research more focused and better suited for generalizability. As with all qualitative research, there are inherent difficulties with replication. The accuracy of the information provided by the participants was dependent on the participants' ability and willingness to be honest in telling their stories. As with all studies, social desirability remains a risk. We cannot exclude the possibility of recall bias. Likewise, the authors had to rely on recruiters from organizations providing social services, police departments, children, parents, and policemen, and because of this, the cases are not representative. The study represents the experiences of ten children, ten parents, ten police officers, and ten CRPS workers from Lithuania (other researchers, however, have corroborated some of their experiences). Due to the limitations of the scope of this article, we were not able to reveal a broader and more nuanced view of the diversity of the experiences of parents, children, and professionals in relation to contextual variables. Nevertheless, this study constitutes an important step forward with regards to documenting the experiences of children, parents, police officers, and CRPS on CRPS's aggression manifestations and preconditions.

## 4.3. Practical Implications

Based on the study results, some implications for practice can be deduced, which can contribute to the improvement of the procedure for removing a child from an unsafe family environment: First, efforts should be made to make it less painful and harmful for children. Second, efforts should be focused on producing more constructive consequences for the entire family. Third, efforts should be made to provide necessary support for CRPS professional practice.

Individuals who were removed from families as children and parents from whom children were removed associated CRPS aggression mainly with micro-level preconditions, such as personal characteristics, motivation, and competencies of specialists. Therefore, it is essential to consider the qualifications, competencies, values, work, and life experience of the candidates when selecting specialists.

The CRPS themselves and police officers primarily associated the manifestations of specialists' aggression with their reaction to emotionally strongly "charged" situations in families and clients' misbehavior. On the one hand, the performance of professionals in

such circumstances requires a very high level of professionalism so that their behavior does not further provoke clients. On the other hand, they have to be able to manage complex and often aggressive situations arising from the unstable emotional state of clients. Thus, it is necessary to take care of CRPS-adapted training and constant supervision to create a psychological support system.

Feedback from front line workers and clients would help develop a supportive institutional environment and relationship culture (including both vertical and horizontal relationships) and improve the work tools. Improved teamwork and established communication procedures with managers would allow professionals to receive more feedback and share responsibilities in dealing with highly complex cases. Greater involvement of clients in decision making related to their well-being would help foster a more empowering perspective and reduce threats related to power relationships [55].

Changes in the organization of family services could also contribute to the professional functioning of the CRPS in specific situations. When organizing the provision of services, it is essential to (a) enable professionals to have enough time to work directly with clients and free them from administrative tasks; (b) seek preventive and continuous work with clients; (c) develop comprehensive but non-restrictive work procedures; and (d) consistently coordinate and constantly reflect on the experience of cooperation with specialists from other institutions involved in taking a child from an unsafe family environment. As to a large extent, inter-institutional cooperation and political and societal preconditions are factors that cause the CRPS to malfunction in the faulty system and environment; the process that aims to improve them would be a complex initiative that requires changes in many institutional structures and processes.

### 4.4. Further Research

In future research, researchers should examine essential aspects of the problem of CRPS aggression towards clients and, notably, use more extensive and more representative samples and even quantitative research design. It should be clarified to what extent the issue addressed in this article is widespread and how it depends on various contextual circumstances or variables. Those variables include the (age of the removed children, their (un)willingness to be taken away, the state of the parents at the time of removal and reasons for removal, the nature of the removal procedure (probability of taking children to the relatives, separation from siblings, duration of removal, and recovery of the child. Of course, international comparative studies conducted in different countries with different CRP systems and procedures would be relevant in this case.

### 5. Conclusions

The results of the study allow us to draw three main conclusions. First, looking at the manifestations and preconditions of aggression through the perspectives of four types of actors helps us address the issue of reliability of information about employee aggression highlighted by other researchers who argue that employees are more likely to not fully disclose their aggressive actions to clients for fear of being misunderstood or hurt. Second, the analysis of the problem of aggression through the perspectives of four types of actors reveals a holistic, multi-layered picture of the problem and allows us to see the broader context of the situation and possible links between these behaviors during child removal. In turn, this encourages a focus on employee accusations of lack of competence or inability to work with clients as the cause of aggression. Third, based on the study's results, suggestions can be made on how the procedure of removing a child from an unsafe family environment can be improved: Firstly, by making it less painful and harmful for children; secondly, to foresee more constructive consequences for the entire family and to make this procedure more empowering; and thirdly, to make the CRP system operate in such a manner that it does not create preconditions for CRPS to transgress the boundaries of professional relations.

**Author Contributions:** Conceptualisation and investigation, D.P., V.G., J.C., M.S., L.Ž., E.Š. and L.G.; methodology, D.P., V.G. and L.Ž.; data curation, D.P. and M.S.; formal analysis, D.P., M.S. and J.C.; writing—original draft, D.P., V.G., J.C. and M.S.; writing—review and editing, L.Ž., E.Š. and L.G.; visualization—M.S.; project administration, L.Ž., D.P. and J.C. All authors have read and agreed to the published version of the manuscript.

**Funding:** This research was supported by the Research Council of Lithuania (LMTLT) under Grant No. S-MIP-19-37. 2019–2021.

**Institutional Review Board Statement:** Not applicable.

**Informed Consent Statement:** Not applicable.

**Data Availability Statement:** Not applicable.

**Acknowledgments:** The authors express gratitude to Elena Cabiati from the Catholic University of Sacred Heart in Milan whose guidance and corrections of the draft version were beneficial and encouraging.

**Conflicts of Interest:** The authors declare no conflict of interest.

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
