# Peer review of "Manifestations and Preconditions of Child Rights Protection—Specialists’ Aggression towards Caregivers and Child in the Situation of Child Removal from the Family"

_sustainability, doi:10.3390/su132011276_

Round 1

Reviewer 1 Report

I would like to thank the authors for an interesting paper. The qualitative methodology of Manifestations and Preconditions of Child Rights Protection Specialists’ Aggression towards Clients was quite strong, and I particularly appreciated the multi-perspective approach which included the voices of all stakeholders. I am aware of research that focuses on the voices of the parents, the voices of children (both as adults but also as children), the voices of child protection/social workers, but less so that of the police and even less so having them all combined and playing off one another within one study like this. I also appreciated very much the idea of  framing this as ‘pre-conditions’ which in a sense provides a potential roadmap as to why child removal so often goes very wrong before, rather than after, the fact.

There is no fatal flaw that I can see that would prevent this paper from being published or necessitating a radical revision. The authors have been very careful in presenting and explaining the methodology and the ethical review process, as well as the limitations of the research. In terms of the actual qualitative findings, there is nothing here that I suspect will surprise anyone who has worked in the area of child protection either as a professional or an academic researcher (or both). On the one hand, that speaks to the strength of the research. I am not familiar with the Lithuanian context in specific, but many of the findings are familiar to me in my own research context as one would expect as child protection cases concerning custody deprivation concern many similar issues internationally. However, I think the idea of thinking about the number of ‘pre-conditions’ leading to this aggression is an interesting way to think about it, and as such this paper should be an interesting inclusion into the debate and worthy of publication.

I do, however, want to spend a moment to talk about the concept of ‘aggression.’ I do think it works and while I applaud the authors for turning the lens of aggression toward the child protection workers (and system they operate within), I think some care needs to be taken with certain statements. For example, the statement “However, research on the aggressive behavior of care professionals working in the field of child rights protection (further CRP) is scarce.” I think this statement is a little too strong. If this is in reference to research in Lithuania, maybe, but I don’t know the Lithuanian language work. If this is in reference to the international English literature, I don’t think ‘scarce’ is a term I would use. ‘Limited’ might be a better term. In my field, child protection issues pertaining to disabled parents in specific, there is a somewhat growing body of work in this area of aggressive, hostile and arguably ‘violent’ attitudes and practices by workers within child protection since at least the 1990s. This is mainly work done in the UK, the Nordic countries, North America and Australia in particular – so if the authors lack a background in disability issues and child protection, overlooking this work may be understandable. While it is true that the term ‘aggression’ is not used per se, Timothy and Wendy Booth’s (UK) well-known thesis of ‘system abuse’ used to describe the unfair and unjust removal of children from disabled parents discusses many of the same issues, and other scholars have followed their lead, such as Llewellyn and McConnell and Spencer (Australia); Aunos and Pacheco (Canada); and Sigurjónsdóttir (Iceland), just to name a few. This work ranges from the 1990s up to the present and uses concepts like ‘system abuse’ and ‘structural violence’ to describe pretty much the same thing – ranging from the poor practices of individual workers, to system level abuses, to cultural/social issues like stigma and prejudice on the part of workers toward parents. To be fair to the authors, perhaps they did not consider the somewhat narrow focus of disabled parents and child protection in specific, but if they did there is no lack of a focus on the aggressive attitudes of child protection workers in this body of work. But this does not really change the authors’ point that this focus on aggression or violence on the part of the workers (or the system) is less-common, but to note that this body of work does exist. However, I would add that the concept of ‘aggression’ works fine, and whether or not we use ‘aggression’ or ‘system abuse’ or ‘structural violence,’ we are talking about some very similar things that seem to extend across countries. It is good to see another country, here Lithuania, added to this and on my part would welcome them into the discussion.

I have a number of comments of lesser importance as well – they are just there for the authors to take or leave as they wish. I agree with the authors that child protection workers are often unfairly targeted in the media, as are often parents too. I think this speaks to the age-old ‘pendulum’ problem in social work in this area: if child protection workers are not aggressive enough with custody deprivation, children may be placed and risk of injury or death leading to criticisms that child protection did not ‘do enough’ and are seen as weak and incompetent. If they are too aggressive and remove children unfairly, then child protection are criticised for being too aggressive and cast in the villain role. As such, when you are criticised no matter what you do it can be frustrating. I can agree with the statement: “Not surprisingly, help professionals avoid openly sharing their experiences or are unwilling to discuss topics of aggression.” This is a drawback of this kind of research, as it makes it harder to explore the issues with this silence. However, I would suggest the authors think (not necessarily for this paper) about this in another way. Avoiding talking about aggression or violence in this context may be done for this reason. In my experience, child protection workers are often shocked (and sometimes angry) when myself and my colleagues discuss their working methods, and outcomes, as ‘violent.’ The issue is that some never conceive of what they are doing as unjust, aggressive, unprofessional or even violent – they are just ‘protecting kids.’ But when you analyse their work in the context of marginalised parents (whether based on class, disability, immigrant or ethnic minority status), it is indeed often aggressive and violent (whether by intent or by outcome). The problem is that some never think that what they are doing is aggressive or violent or flawed, so they really have a hard time speaking to the issues. Some may be unwilling to talk about this for the reasons the authors suggest, yes, but some may be unwilling because they really don’t see that they are doing anything wrong in the first place. In my experience this is true of disability because their working methods are based on the idea that (certain) disabled parents should not be parents in the first place, and all their decisions are informed by that view – so whatever shortcuts or liberties or less than ideal practices they are engaged in are justified by the ‘rescuing kids from bad parents’ perspective. I would imagine that this extends to Lithuania as well in the context of disability – you also see this with other oppressed groups, such as indigenous groups in Australia and North America – that they are not ‘fit’ parents on the basis of who they are. In my experience some child protection workers always see child removal as the only just course of action for some ‘kinds’ of parents and would never conceive of what they are doing as aggressive or violent, or else find various rationalisations or justifications for it. So there may be multiple reasons for ‘not talking about it.’

Section 3.5: Preconditions of CRPS aggression towards clients: children’s perspective. “some children attributed their difficult experiences at the time of their removal to the fact that CRPS performed their duties in a formal manner, and some believed that it was done on the basis of material gain: “it feels as if they work only because they get money for it”. For me there is something a little off here, and perhaps unfair toward child protection workers. I understand the criticism of the ‘formal manner’ and I am sure some act this way because they believe it to be professional, or as a distancing measure (to maintain some sense of objectivity in a difficult job). And I could see why a child would be disturbed by this. But I find the quote here more confusing than helpful – of course child protection workers are doing it for ‘money’ as it is their job and this is what they do, and they need to be paid to eat and pay for housing. The same holds for doctors, factory workers and uni professors, you name it. We all need to eat so we need to get paid. I don’t think I would do my job as a professor if I did not get paid either. I don’t think being a child protection worker is a calling, like being a monk or a nun, so it seems a little unfair to use this against them. Here we are not actually talking about children, but adults interviewed about their experiences as children, so they would understand this as well. So it is a little unclear to me. Could the criticism be that the problem is that they are treating their job just like any other? In other words, if I cannot find something at a grocery store, I expect help, but I don’t expect the clerk (often an underpaid immigrant or uni student in my country) to bend over backwards to help either. However, in a very sensitive environment like child protection interventions, could it be the criticism is that they were doing their work ‘like a job,’ in other words, in a dispassionate and uninterested way, like a grocery store clerk? That criticism would make more sense to me, as I would also expect a doctor or therapists or anyone in the care or protection services to be a little more engaged as well. But I would not hold it against a doctor or social worker to be ‘paid’ per se, which is my point where it seems like an unfair criticism and I am wondering if it is more that your informant was trying to criticise that they are doing this sensitive work ‘just like any other work,’ rather than the ‘pay’ issue per se? I cannot see ‘being paid’ as a precondition for aggression, but I can accept the argument that if workers are treating this job just like making hamburgers or pumping gas, that can be problematic. And this does make me think how many child protection workers I have encountered are not well-suited to this kind of work.

3.7. Preconditions of CRPS aggression towards clients: police officers’ perspective: There is a quote from an interview here which says (“there are people who look outright”). I suspect there is a Lithuanian to English translation issue here, as I have no idea what it means that people look ‘outright.’ It is about police officers who based judgements on appearances, I think. I don’t know if there are any immigrant groups in Lithuania or Roma people, so maybe this is in relation to class? In other words, maybe people who ‘look off’ or ‘look marginalised’ or something like that? I suspect the cop meant people who appear underclass or marginalised, like what people in the UK call ‘chavs’ or something like that – in other words, people who the police expect to be violent, or use drugs, or whatnot based on appearance and they treat them as such, even if the people in question are nothing like that. I have a feeling this is what this statement is about, but I am not sure, except that the word ‘outright’ is not the right word here.

Line 471 – another language issue. The phrase “media agiotage.” I never encountered the word agiotage until this day – I looked it up and it is a specialist business term about the buying and selling of stocks, so I suspect the wrong word was chosen here. 

Author Response

Thank you for your very valuable comments and suggestions. Please, see our responses and changes we made in attached file.

Reviewer 2 Report

Very relevant article. Well developed introduction of the methodology and well developed argumentation.  The quality of the article can be improved with a deeper discussion of the limits to generalize from the results of the collected data .

Author Response

(The authors gave the same response as above.)

Reviewer 3 Report

Dear authors,

The manuscript focuses on an interesting and relevant topic of research. I have appreciated your contributions and work and valued the effort.

Considering the nature of the journal, I suggest creating a relationship with sustainability. 

If the study is focused on issues inherent to taking a child from home/ parents, perhaps the title should be reviewed to be more informative. 

The introduction would benefit from adding references, for example, lines 46-54. More knowledge about specialists practices is needed, not just the impact of it.

In the methodology, identify literature/ authors that support your options and their execution in the field. Give examples of categories that emerged. How was the analysis, what models/ techniques/ strategies?   

The results presentation is somehow confusing:

-Please identify the participant in each quote, within an anonymous format. Edition of the quotes should be uniformized, for example, adding quotation marks (e.g. lines 181-183 or lines 356-359).

-Could results be organized according for example the emerged categories instead of participants' perspectives? And systematize it more clearly. 

-In results about "Preconditions", four categories (?) are listed for each participant group's perspective. Clarify how they emerged. Should they be defined and discussed according to the literature in the introduction part? 

Hope my comments contribute to the paper improvment.

All the best.

Author Response

(The authors gave the same response as above.)

Round 2

Reviewer 3 Report

Dear authors 

Improvements value the paper.